# Characterization of Sweet Corn Production in Subtropical Environmental Conditions

Jessica Paranhos [1], Wheeler Foshee [1], Timothy Coolong [2], Brian Heyes [3], Melba Salazar-Gutierrez [1], Kathelyn Kesheimer [4] and Andre Luiz Biscaia Ribeiro da Silva [1,*]

[1]  101 Funchess Hall, Department of Horticulture, Auburn University, Auburn, AL 36849, USA
[2]  1111 Miller Plant Sciences, Department of Horticulture, University of Georgia, Athens, GA 30602, USA; tcoolong@uga.edu
[3]  Mitchell County Extension, University of Georgia Cooperative Extension, Camila, GA 31730, USA
[4]  105 Extension Hall, Department of Entomology and Plant Pathology, Auburn University, Auburn, AL 36849, USA
*   Correspondence: azb0207@auburn.edu

**Abstract:** Weather variability in subtropical environmental conditions of the southeastern U.S. impact sweet corn production in the region, which is one of the most important in the country. Understanding sweet corn performance under these environmental conditions is important to help growers with decision making. Thus, the objectives of this study were to evaluate and characterize the performance of ten commercial sweet corn cultivars exposed to several environmental conditions of the southeastern U.S. and to describe impacts of weather variability on cultivar development, yield, and ear quality. Field experiments were conducted in five locations of the southeastern U.S. during the spring and fall of 2020 and 2021. Weather data, biomass accumulation, yield, and ear quality were measured for all cultivar within seasons and locations. Heavy rainfall events created waterlogging conditions for sweet corn development; however, it was the daily air temperature of seasons that mostly impacted yield and ear quality. Daily air temperatures extended the growing season of spring but reduced crop development in the fall. Consequently, biomass accumulation was generally higher in the spring (4243 kg ha$^{-1}$) compared to the fall (1987 kg ha$^{-1}$). Biomass accumulation translated into yield, which was thereby higher in the spring compared to the fall. Cultivars with great potential against environmental stresses and best performance for most locations were Affection, GSS1170, Passion, and SCI336 in the spring, and Affection, GSS1170, and SC1136 in the fall. Ultimately, sweet corn yield was strongly correlated with ear dimensions but poorly correlated with number of grains in a kernel, suggesting that breeding programs trying to increase potential yield should be focused on ear diameter and length.

**Keywords:** weather variability; yield; ear parameters; multivariate analysis

## 1. Introduction

Sweet corn (*Zea mays* subsp. *Mays* L.) is an annual grass and a warm-weather vegetable crop widely grown in the U.S., where it ranks the third-most-grown vegetable crop [1–3]. Annually planted in approximately 150,178 ha, sweet corn is valued at USD 775 million, with 44% of the national production relying on the environmental conditions of the southeastern U.S. [4].

The southeastern U.S. is classified with a humid subtropical climate (Cfa), characterized by heavy rainfall events during summer and dry periods during winter [5,6]. This climate is considered optimal for sweet corn production; however, the recent high spatial and temporal variability of regional weather conditions, also known as weather variability, has created challenges for sweet corn production in the southeastern U.S. [5,7,8]. Daily air temperatures have been impacting seed germination, root and leaf development, tasseling, pollination, grain filling, and yield [8–10]. Particularly, sweet corn yield has been reported

to drop by 23% due to heat stress in the southeastern U.S. [7]. Changes in rainfall patters have also been creating challenges [11]. Heat and drought events were reported to increase osmotic stress and reduce seed germination, plant growth, leaf expansion, and ear development [12]. When heat stress is present during ear differentiation, there is a decrease in ear length and the number of kernel rows. When heat stress is present during tasseling, there is a significant reduction in ear weight [13]. In the case of excessive rainfall events, saturated soils are reported to cause anaerobic conditions in the root zone, reducing water uptake, stomal conductance, photosynthesis rate, and chlorophyll content [13]; ultimately, excessive rainfall events reduce grain fill and ear weight [13].

Overall, the current weather variability of the subtropical climate in the southeastern U.S. requires a better understanding of how weather is affecting the sweet corn growing seasons, plant development, yield, and ear quality. Information can help growers to ensure crop quality and potential yields. Thus, the objectives of this study were to evaluate and characterize the performance of ten commercial sweet corn cultivars exposed to several environmental conditions of the southeastern U.S. and to describe the impacts of weather variability on cultivar development, yield, and ear quality.

## 2. Materials and Methods

### 2.1. Sites Description and Experimental Design

Field experiments were conducted in collaboration with sweet corn growers during the spring and fall growing seasons of 2020 at three sites in Georgia, U.S., and during the spring and fall growing season of 2021 at two sites in Alabama, U.S. (Table 1). All five locations were classified within the humid subtropical climate (Cfa), with heavy rainfall events during a hot summer and dry periods during the winter [5,6]. The soil characteristics of each location are shown in Table 1.

**Table 1.** Location, geographic coordinates, year, season, soil type, planting space (IRS), planting date (PD), biomass sampling events (S) in days after planting (DAP), harvesting date, and growing degree days (GDD) accumulated for all field experiments.

| Location | Coordinates | Year | Season | Soil Type | IRS (cm) | PD | S0 | S1 | S2 | S3 | S4 | S5 | Harvest | GDD |
|---|---|---|---|---|---|---|---|---|---|---|---|---|---|---|
| Southwest GA | 31.18269° N 84.40958° W | 2020 | Spring | Troup sand | 15.24 | 15 April | 1 | 43 | 55 | 69 | - | - | 69 | 898 |
| | | 2020 | Fall | | 17.78 | 26 August | 1 | 14 | 28 | 47 | 66 | - | 66 | 928 |
| Southeast GA | 32.01807° N 82.22108° W | 2020 | Spring | Irvington loamy sand | 15.24 | 3 June | 1 | 14 | 30 | 44 | 58 | - | 58 | 930 |
| | | 2020 | Fall | | 17.78 | 21 August | 1 | 19 | 40 | 54 | 68 | - | 68 | 898 |
| South GA | 31.42378° N 83.68807° W | 2020 | Spring | Tifton loamy sand | 15.24 | 2 April | 1 | 47 | 60 | 68 | - | - | 68 | 877 |
| | | 2020 | Fall | | 17.78 | 18 August | 1 | 15 | 31 | 44 | 57 | 66 | 66 | 916 |
| Southwest AL | 31.14055° N 87.04885° W | 2021 | Spring | Benndale fine sandy loam | 15.24 | 23 March | 1 | 52 | 65 | 85 | - | - | 85 | 930 |
| | | 2021 | Fall | | 15.24 | 6 August | 1 | 19 | 40 | 63 | 73 | - | 73 | 992 |
| Central AL | 32.50058° N 85.89150° W | 2021 | Spring | Kalmia loamy sand | 15.24 | 14 April | 1 | 30 | 43 | 64 | 78 | - | 78 | 991 |
| | | 2021 | Fall | | 15.24 | 12 August | 1 | 29 | 48 | 64 | 78 | - | 78 | 965 |

In all locations, a factorial experimental design of sweet corn cultivars was arranged in a complete randomized block design with four replications in the Georgia sites and three replications in the Alabama sites. Sweet corn cultivars (n = 10) are described in Table 2. Experimental units were comprised of 80 sweet corn plants in all sites. Crop management practices associated with soil preparation, irrigation, and management of pests, weeds, and diseases followed the recommendations of the Southeastern U.S. Vegetable Crop Handbook, for all locations [14].

**Table 2.** Overview of sweet corn commercial cultivars evaluated.

| Cultivar | Color | Disease Resistance |
|---|---|---|
| Passion | Yellow | Rust, HR: Rp1D, IR: Pst, Et |
| SCI336 | Yellow | M: Ps, Et, Pst |
| Obsession | Bicolor | Ps, Et, Pst |
| Affection | Bicolor | - |
| EX08767143 | Bicolor | Rust, IR: Et, Pst |
| Coastal | Bicolor | HR: Ps (Rp1-g) |
| Flagler | Bicolor | HR: Ps (Rp1-g) |
| BSS1075 | Bicolor | HR: PS: Rp1-i |
| BSS8021 | Bicolor | HR: PS: Rp1-i, Et |
| GSS1170 | Yellow | HR: Et, Ps: Rsp1-i |

HR: high resistance; M: moderate resistance; IR: intermediate resistance; Rp1D, Rp1-g, Rsp1-i, and Rp1-i: genes that confer resistance to *Puccinia sorghi*, agent of common rust; Ps: fungus *Puccinia sorghi* (common rust); Pst: bacteria *Pantoea stewartii* (Stewart's wilt); Et: fungus *Exserohilum turcicum* (northern leaf blight).

### 2.2. Weather Data and Growing Degree Days (GDD)

During all growing seasons, the daily maximum and minimum air temperature and rainfall events in each location were monitored using the closest weather station—either the Georgia Automated Weather Network or the Auburn University Mesonet.

Accumulated growing degree days (*GDD*) were determined using the following equation.

$$GDD = \frac{(Tmax + Tmin)}{2} - Tbase$$

where "*Tmax*" means average daily maximum temperature, "*Tmin*" means average daily minimum temperature, and "*Tbase*" means the sweet corn base temperature, which was set at 10 °C [7].

### 2.3. Biomass Accumulation, Yield, and Ear Quality

Sweet corn biomass was monitored with plant tissue samples collected at least four times during each growing season (Table 1). Samples were comprised of two representative plants of each plot, oven-dried at 65.5 °C until they reached a constant weight. Subsequently, the sweet corn maximum crop biomass accumulation (NM), sweet corn biomass accumulation rate constant (*k*), and half maximum sweet corn biomass accumulation (*l*) of each variety within each season and location were simulated by fitting sweet corn biomass data into the Witty (1983) model [15] using the Sigma Plot Version 14.5 (Systat Software), as follows:

$$Crop\ biomass\ accumulation = \frac{NM}{1 + e^{-k(t-l)}}$$

where "NM" is maximum crop biomass accumulation, "*k*" is crop biomass accumulation rate constant, "*t*" is time in days, and "*l*" is days to half maximum biomass accumulation.

At maturity, sweet corn ears were harvested in all locations (Table 1). During harvest, the number of ears and total weight were recorded. Additionally, five ears were randomly selected from each plot and ear length, ear diameter, number of kernel rows in an ear (KR), number of kernel grains in an ear row (KIR), and the total number of kernels in an ear (KTG) were measured.

### 2.4. Statistical Analysis

Statistical analyses were performed using linear mixed techniques as implemented in the SAS PROC GLIMMIX procedure (SAS/STAT 9.4; SAS Institute Inc., Cary, NC, USA). All response variables were analyzed with location, year, and season as fixed effects. Blocks within each location and season were considered a random effect. When the F value of the analysis of variance was significant, least-square means comparisons were performed using Tukey's Honest Significant Difference Test ($p < 0.05$), and means were portioned using the slice command in SAS.

A multivariate analysis was also performed using the R Studio software (version 4.0.2), RStudio (RStudio Team 2020, Boston, MA, USA). The dissimilarity among all response variables (biomass, yield, and ear quality parameters) was measured by the Euclidean distance and presented as a cluster analysis, which was built based on a hierarchical unweighted pair-group method using arithmetic averages. In addition, all data were submitted to a principal component analysis (PCA) to verify the contribution of biomass, yield, ear quality parameters, cultivars, location, and season to the construction of the principal components. A correlation-based network analysis was also performed using Pearson's method.

## 3. Results

### *3.1. Weather Data and Growing Degree Days (GDD)*

Rainfall events and the minimum and maximum daily air temperature of all locations are shown in Figure 1, while the total GDD accumulated within each season of all locations is shown in Table 1.

In southwest GA, average minimum and maximum daily air temperatures were 17 and 30 °C during the spring growing season and averaged 19 and 30 °C during the fall growing season of 2020, respectively. There was 280 and 233 mm of precipitation in spring and fall 2020, respectively.

In southeast GA, average minimum and maximum daily air temperatures were 22 and 33 °C during the spring season and averaged 18 and 28 °C during the fall growing season of 2020, respectively. There was 218 and 175 mm of precipitation in the spring and fall 2020, respectively.

In south GA, average minimum and maximum daily air temperatures were 15 and 28 °C during the spring growing season and averaged 18 and 28 °C in the fall growing season of 2020, respectively. There was 150 and 229 mm of precipitation at the study locations in spring and fall 2020, respectively.

In southwest AL, average minimum and maximum daily air temperatures were 15 and 27 °C during spring 2021 and averaged 19 and 29 °C in the fall 2021, respectively. There was 446 and 355 mm of precipitation at the study locations in spring and fall 2021, respectively.

In central AL, average minimum and maximum air daily temperatures were 17 and 29 °C during the spring growing season and averaged 18 and 29 °C in the fall growing season of 2021, respectively. There was 303 and 293 mm of precipitation at the study locations in spring and fall 2021, respectively.

### *3.2. Biomass Accumulation*

Biomass accumulation was not statistically compared among cultivars, locations, and seasons. Instead, biomass accumulation was fitted in the Witty (1983) model for the characterization of the performance of sweet corn cultivars [15]. Table 3 displays the NM, $l$, and $k$ values for all cultivars within each season and location. In general, the NMs of sweet corn cultivars were greater in the spring season compared to fall season within all locations. However, $l$ and $k$, which indicate sweet corn growth, were greater for the fall season compared to the spring season.

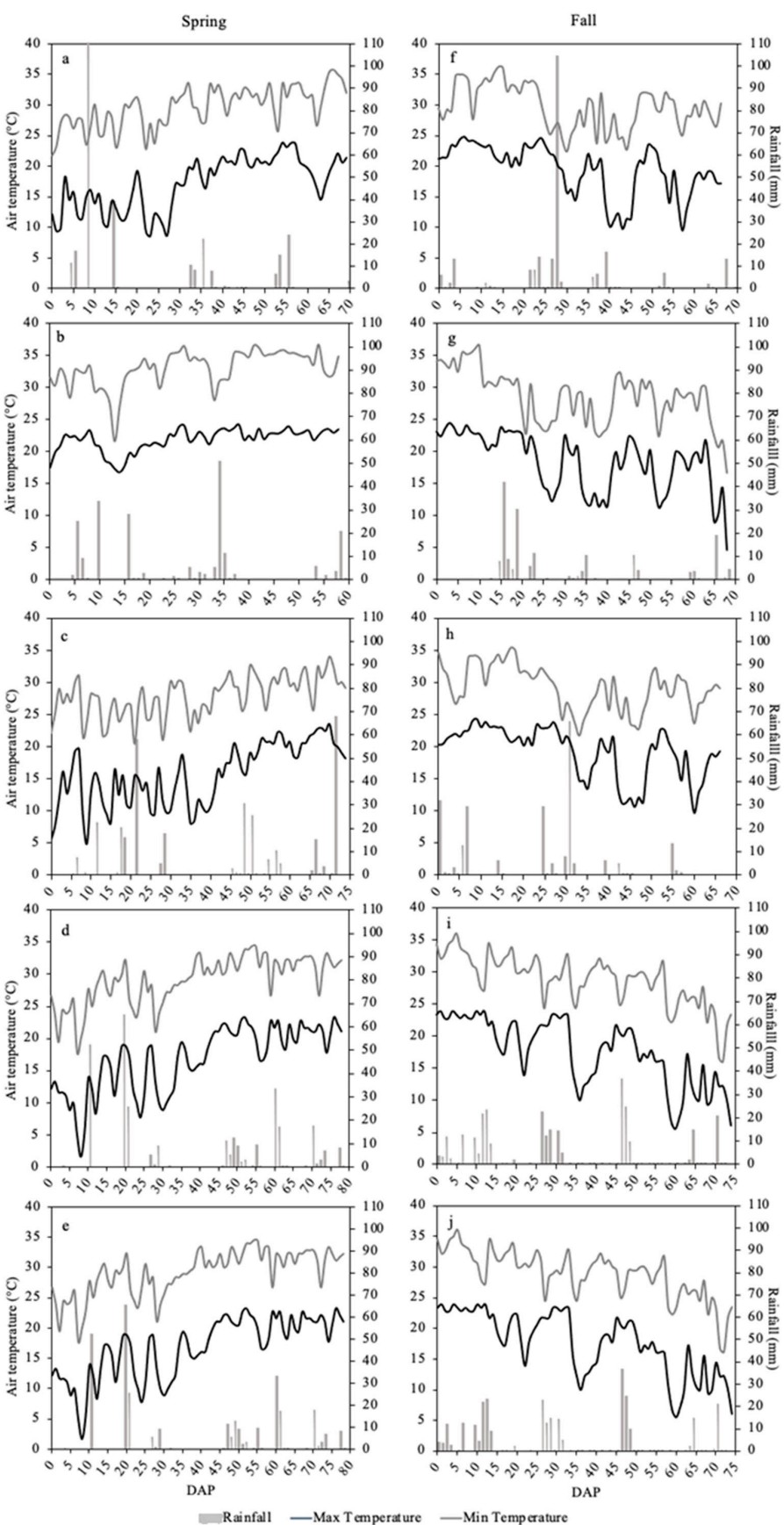

**Figure 1.** Rainfall and maximum and minimum daily air temperature in the spring (**a–e**) and fall (**f–j**) of southwest GA (**a,f**), southeast GA (**b,g**), south GA (**c,h**), southwest AL (**d,i**), and central AL (**e,j**).

**Table 3.** Effect of cultivar, location, and season on sweet corn maximum biomass accumulation (NM), days to reach half biomass (*l*), and crop biomass accumulation rate (*k*).

| Cultivar | Southwest GA | | Southeast GA | | South GA | | Southwest AL | | Central AL | |
|---|---|---|---|---|---|---|---|---|---|---|
| | Spring | Fall | Spring | Fall | Spring | Fall | Spring | Fall | Spring | Fall |
| NM (kg ha$^{-1}$) | | | | | | | | | | |
| Affection | 3505 | 1663 | 4030 | 2293 | 2819 | 1833 | 456 | 1418 | 9448 | 1754 |
| BSS1075 | 3880 | 2127 | 5946 | 2638 | 2759 | 2137 | 1833 | 1678 | 8372 | 2671 |
| BSS8021 | 3418 | 1947 | 4971 | 1777 | 2762 | 2209 | 486 | 1739 | 7056 | 1880 |
| Coastal | 3280 | 1749 | 3307 | 2460 | 3029 | 2010 | 604 | 1009 | 8731 | 2207 |
| EX08767143 | 3475 | 1884 | 4641 | 2471 | 2726 | 2190 | 1237 | 1327 | 9568 | 2232 |
| Flagler | 3433 | 2084 | 4263 | 2474 | 3193 | 2266 | 1167 | 1610 | 9807 | 1798 |
| GSS1170 | 3534 | 2191 | 4577 | 2221 | 2656 | 2219 | 506 | 1103 | 9149 | 1862 |
| Obsession | 3522 | 1784 | 5409 | 2546 | 2721 | 2073 | 1053 | 1731 | 10525 | 2107 |
| Passion | 3851 | 1959 | 5468 | 2546 | 2394 | 2189 | 879 | 1391 | 8731 | 2043 |
| SCI336 | 3506 | 2262 | 5115 | 2456 | 2457 | 2179 | 1150 | 1338 | 10764 | 1618 |
| *l* (days) | | | | | | | | | | |
| Affection | 43.2 | 29.2 | 37.8 | 41.2 | 47.3 | 31.8 | 51.7 | - | 46.2 | 29.8 |
| BSS1075 | 42.9 | 29.3 | 38.7 | 39.8 | 48.7 | 34.1 | 52.6 | - | 46.4 | 29.6 |
| BSS8021 | 42.6 | 29.6 | 38.5 | 15.2 | 51.6 | 33.7 | 52.1 | - | 46.0 | 29.6 |
| Coastal | 42.7 | 29.0 | 35.9 | 34.0 | 47.1 | 34.5 | 52.1 | - | 46.0 | 30.0 |
| EX08767143 | 42.7 | 29.0 | 36.2 | 36.8 | 45.7 | 37.0 | 52.3 | - | 46.4 | 29.9 |
| Flagler | 42.9 | 30.9 | 38.2 | 39.7 | 51.5 | 36.5 | 52.6 | - | 46.2 | 3.00 |
| GSS1170 | 43.3 | 35.3 | 37.3 | 40.0 | 47.7 | 35.5 | 51.8 | - | 46.1 | 29.6 |
| Obsession | 42.9 | 32.2 | 39.0 | 39.7 | 49.3 | 33.6 | 52.6 | - | 46.3 | 30.1 |
| Passion | 41.7 | 29.1 | 38.5 | 38.2 | 51.0 | 34.6 | 52.0 | - | 46.3 | 29.6 |
| SCI336 | 40.4 | 37.7 | 39.0 | 40.4 | 45.3 | 36.3 | 54.6 | - | 46.4 | 30.3 |
| *k* | | | | | | | | | | |
| Affection | 1.681 | 0.939 | 0.203 | 0.199 | 0.203 | 1.293 | 1.557 | 1.392 | 1.159 | 1.176 |
| BSS1075 | 1.703 | 0.721 | 0.148 | 1.509 | 0.241 | 0.276 | 1.621 | 1.427 | 1.184 | 1.086 |
| BSS8021 | 1.719 | 0.648 | 0.15 | 0.298 | 0.126 | 0.163 | 1.735 | 1.432 | 1.129 | 1.095 |
| Coastal | 1.622 | 0.808 | 0.187 | 0.079 | 0.133 | 0.227 | 1.612 | 0.429 | 1.173 | 1.111 |
| EX08767143 | 1.325 | 0.6 | 0.202 | 0.117 | 0.167 | 0.141 | 2.604 | 1.423 | 1.172 | 1.193 |
| Flagler | 1.612 | 0.265 | 0.133 | 0.213 | 0.125 | 0.237 | 1.529 | 0.062 | 1.172 | 1.188 |
| GSS1170 | 1.79 | 0.183 | 0.184 | 0.225 | 0.244 | 0.177 | 0.051 | 1.323 | 1.138 | 1.126 |
| Obsession | 1.817 | 0.227 | 0.173 | 1.437 | 0.134 | 0.35 | 1.661 | 1.413 | 1.247 | 1.088 |
| Passion | 0.209 | 0.682 | 0.152 | 0.203 | 0.138 | 0.236 | - | 1.356 | 1.192 | 2.069 |
| SCI336 | 0.206 | 0.142 | 0.158 | 0.229 | 0.215 | 0.161 | 0.395 | 1.378 | 1.198 | 1.105 |

### 3.3. Sweet Corn Yield

Sweet corn yield was significantly impacted by the interaction among cultivars, locations, and seasons (Table 4).

In southwest GA, cultivars Obsession (24.3 Mg ha$^{-1}$) and Passion (24.3 Mg ha$^{-1}$) had the highest yields in the spring, while cultivars GSS1170 (27.6 Mg ha$^{-1}$) and Affection (25 Mg ha$^{-1}$) had the highest yields in the fall. In southeast GA, cultivars EX08767143 (26.7 Mg ha$^{-1}$) and Coastal (23.7 Mg ha$^{-1}$) had the highest yield in the spring, while cultivars Affection (28.2 Mg ha$^{-1}$) and Coastal (27.8 Mgha$^{-1}$) had the highest yields in the fall. In south GA, cultivars Coastal (19.1 Mg ha$^{-1}$), Affection (17.7 Mg ha$^{-1}$) and GSS1170 (17.7 Mg ha$^{-1}$) had the highest yields in the spring, while cultivars Affection (32.1 Mg ha$^{-1}$) and SCI336 (30.5 Mg ha$^{-1}$) had the highest yields in the fall. In southwest AL, cultivars Coastal (35.8 Mg ha$^{-1}$) and BSS1075 (23.9 Mg ha$^{-1}$) had the highest yields in the spring, while cultivars SCI336 (15 Mg ha$^{-1}$) and GSS1170 (14.8 Mg ha$^{-1}$) had the highest yields in the fall. In central AL, cultivars EX08767143 (26.5 Mg ha$^{-1}$) and SCI336 (25.7 Mg ha$^{-1}$) had the highest yields in the spring, while cultivars EX08767143 (19.7 Mg ha$^{-1}$) and BSS1075 (16.7 Mg ha$^{-1}$) had the highest yields in the fall.

**Table 4.** Effect of the interaction among sweet corn cultivars, seasons, and locations on sweet corn total yield.

| Cultivar | Southwest GA | | Southeast GA | | South GA | | Southwest AL | | Central AL | |
|---|---|---|---|---|---|---|---|---|---|---|
| | Spring | Fall | Spring | Fall | Spring | Fall | Spring | Fall | Spring | Fall |
| | Mg ha$^{-1}$ | | | | | | | | | |
| Affection | 22.8 ± 1.7 [x] B [y] a [z] | 25.0 ± 1.2 Aab | 18.7 ± 2.5 Bbc | 28.2 ± 2.2 Aa | 17.7 ± 2.2 Bab | 32.1 ± 4.6 Aa | 20.4 ± 5.9 Abcd | 12.3 ± 1.3 Ba | 25.2 ± 2.7 Aa | 16.4 ± 0.3 Bab |
| BSS1075 | 21.3 ± 0.7 Aa | 19.6 ± 0.2 Ac | 20.1 ± 0.2 Abc | 22.3 ± 0.7 Abcd | 16.4 ± 0.8 Bab | 19.1 ± 0.6 Ae | 23.9 ± 1.5 Aab | 8.5 ± 2.6 Ba | 23.1 ± 2.0 Aab | 16.7 ± 0.4 Bab |
| BSS8021 | 22.1 ± 0.8 Aa | 21.3 ± 1.8 Bbc | 16.3 ± 1.8 Ac | 19.1 ± 2.0 Ad | 14.2 ± 1.1 Bab | 19.6 ± 0.7 Ae | 23.1 ± 2.7 Aab | 9.5 ± 0.8 Ba | 19.2 ± 3.0 Ab | 8.7 ± 0.6 Bc |
| Coastal | 19.9± 1.6 Ba | 22.1 ± 0.5 Abc | 23.7 ± 3.7 Bab | 27.8 ± 1.9 Aab | 19.1 ± 2.0 Ba | 25.7 ± 1.1 Abcd | 35.8 ± 1.2 Aa | 9.5 ± 0.3 Ba | 24.8 ± 2.5 Aab | 12.1 ± 0.8 Bbc |
| EX08767143 | 23.2 ± 1.7 Aa | 22.2 ± 0.4 Abc | 26.7 ± 2.5 Aa | 25.4 ± 0.6 Babc | 17.2 ± 1.0 Bab | 26.1 ± 1.2 Abc | 19.7 ± 2.4 Abcd | 11.7 ± 0.7 Ba | 26.5 ± 3.3 Aa | 19.7 ± 0.8 Ba |
| Flagler | 22.0 ± 1.1 Aa | 20.6 ± 0.5 Abc | 19.9 ± 0.9 Abc | 23.5 ± 1.0 Aabcd | 17.2 ± 1.3 Bab | 23.8 ± 0.8 Acde | 26.7 ± 0.9 Ab | 12.1 ± 0.2 Ba | 23.7 ± 3.1 Aab | 15.1 ± 0.7 Babc |
| GSS1170 | 22.5 ± 1.1 Ba | 27.6 ± 2.3 Aa | 19.9 ± 2.4 Abc | 24.9 ± 2.1 Aabc | 17.7 ± 1.9 Bab | 25.8 ± 1.1 Abc | 21.1 ± 1.1 Aabcd | 14.8 ± 1.6 Ba | 25.0 ± 2.2 Aab | 11.5 ± 1.2 Bbc |
| Obsession | 24.3 ± 3.1 Aa | 19.6 ± 0.7 Bc | 18.2 ± 2.3 Bc | 24.5 ± 1.3 Aabc | 14.6 ± 2.1 Bab | 20.7 ± 1.6 Ade | | 11.8 ± 0.8 Ba | 23.2 ± 1.0 Aab | 14.9 ± 0.7 Babc |
| Passion | 24.3 ± 0.6 Aa | 20.7 ± 1.2 Bbc | 19.2 ± 1.5 Abc | 21.3 ± 1.5 Acd | 12.7 ± 1.8 Bb | 22.2 ± 1.0 Acde | 20.0 ± 10.8 Abcd | 11.9 ± 0.8 Ba | 23.4 ± 2.0 Aab | 16.4 ± 0.5 Bab |
| SCI336 | 23.2 ± 0.5 Aa | 23.9 ± 0.7 Aabc | 23.5 ± 3.2 Aab | 24.8 ± 3.2 Aabc | 15.0 ± 1.5 Bab | 30.5 ± 2.1 Aab | 16.8 ± 7.5 Acd | 15.0 ± 1.0 Ba | 25.7 ± 2.0 Aa | 21.2 ± 0.1 Aa |

[x] Denotes the mean standard error. [y] Values followed by similar uppercase letters among season (column) within cultivar (row) indicate no significant difference according to the Tukey mean test. [z] Values followed by similar lowercase letters among cultivar (row) within season (column) indicate no significant difference according to the Tukey mean test.

For the yield comparison among cultivars in the spring and fall seasons within each location, the cultivar Affection had the highest yield, which had no significant difference among location within the fall growing season. Cultivars BSS1075 and BSS8021 had the highest yield in the spring of southwest AL and central AL, and in the fall of southeast GA and south GA. The cultivar Coastal had the highest yields in the spring of central AL and southwest AL, and in the fall of southwest GA, southeast GA, and south GA. Cultivars EX08767143 and Flagler performed similarly and had the highest yields in the spring of southwest GA, southeast GA, southwest AL, and central AL, and in the fall of southwest GA and south GA. The cultivar GSS1170 had the highest yields in the spring for southwest AL, central AL, and southeast GA, and in the fall of southwest GA, south GA, and southeast GA. Cultivars Obsession and Passion had the highest yields in the spring of southwest AL, central AL, and southwest GA, and in the fall of southeast GA and south GA. The cultivar SCI336 had similar yield for all locations in both spring and fall seasons, except for the spring in south GA.

### 3.4. Ear Quality Parameters

Among all ear quality parameters (i.e., ear diameter, ear length, KR, KIR, and KTG, there was a significant interaction between cultivar and location for KTG (Table 5); between location and season for ear diameter, KR, KIR, and KTG (Table 6); and between cultivar and season for ear diameter and ear length (Table 7).

**Table 5.** Effect of the interaction between sweet corn cultivar and location for kernel total grains (KTG).

| Cultivar | Southwest GA | | Southeast GA | | South GA | | Southwest AL | | Central AL | |
|---|---|---|---|---|---|---|---|---|---|---|
| Affection | 581 ± 30.4 [x] | A [y] bc [z] | 520 ± 17.3 | ABbc | 504 ± 27.7 | BCc | 444 ± 24.3 | Cb | 574 ± 34.7 | ABabc |
| BSS1075 | 651 ± 25.4 | Aa | 594 ± 17.8 | Aba | 581 ± 18.4 | Bab | 476 ± 53.8 | Cab | 594 ± 24.5 | ABabc |
| BSS8021 | 533 ± 21.6 | Ac | 514 ± 23.1 | Ac | 526 ± 20.7 | Abc | 465 ± 23.1 | Aab | 511 ± 37.2 | Ac |
| Coastal | 552 ± 19.2 | Abc | 470 ± 16.2 | Bc | 571 ± 38.3 | Aab | 528 ± 45.5 | ABa | 576 ± 37.4 | Aabc |
| EX08767143 | 617 ± 43.3 | Aab | 521 ± 24.2 | Bbc | 526 ± 18.4 | Bbc | 491 ± 45.0 | Bab | 627 ± 30.4 | Aa |
| Flagler | 533 ± 21.9 | ABc | 512 ± 27.0 | ABc | 572 ± 23.3 | Aab | 467 ± 37.8 | Bab | 534 ± 31.3 | ABbc |
| GSS1170 | 551 ± 16.1 | ABbc | 588 ± 23.8 | Aab | 602 ± 36.0 | Aa | 487 ± 3.81 | Bab | 574 ± 36.3 | ABabc |
| Obsession | 576 ± 38.1 | Abc | 596 ± 31.4 | Aa | 564 ± 28.5 | Aabc | 478 ± 44.1 | Bab | 581 ± 39.2 | Aabc |
| Passion | 580 ± 30.0 | ABbc | 589 ± 25.9 | ABab | 529 ± 29.4 | BCbc | 471 ± 25.6 | Cab | 640 ± 52.3 | Aa |
| SCI336 | 671 ± 50.1 | Aa | 583 ± 21.9 | Bab | 581 ± 26.2 | Bab | 487 ± 48.9 | Cab | 620 ± 25.0 | Aba |

[x] Denotes the mean standard error. [y] Values followed by similar uppercase letters among location (column) within cultivar (row) indicate no significant difference according to the Tukey mean test. [z] Values followed by similar lowercase letters among cultivars (row) within location (column) indicate no significant difference according to the Tukey mean test.

**Table 6.** The interaction between seasons and locations for ear diameter, kernel rows (KR), kernel grains in a row (KIR), and kernel total grains (KTG).

| Location | Season | | | | | | | |
|---|---|---|---|---|---|---|---|---|
| | Spring | Fall | Spring | Fall | Spring | Fall | Spring | Fall |
| | Ear diameter cm | | Kernel rows # | | Kernel grains in a row # | | Kernel total grains # | |
| Southwest GA | 4.3 ± 0.0 [x] A [y] cd [z] | 4.4 ± 0.0 Abc | 17.7 ± 0.3 Aa | 16.3 ± 0.3 Ba | 35.8 ± 0.3 Abc | 32.9 ± 0.4 Bab | 632 ± 13 Aa | 536 ± 12 Ba |
| Southeast GA | 4.4 ± 0.0 Abc | 4.2 ± 0.0 Bd | 16.4 ± 0.3 Ab | 16.0 ± 0.2 Aab | 33.7 ± 0.5 Ad | 34.2 ± 0.5 Aa | 552 ± 13 Abc | 544 ± 11 Aa |
| South GA | 4.2 ± 0.1 Bd | 4.5 ± 0.0 Ab | 15.9 ± 0.4 Abc | 16.6 ± 0.3 Aab | 37.0 ± 0.5 Ab | 31.9 ± 0.5 Bab | 583 ± 12 Ab | 528 ± 10 Ba |
| Southwest AL | 4.6 ± 0.1 Aab | 4.3 ± 0.0 Bcd | 15.2 ± 0.3 Abc | 15.4 ± 0.2 Ab | 35.3 ± 0.7 Ac | 27.9 ± 0.6 Bc | 534 ± 15 Ac | 428 ± 11 Bb |
| Central AL | 4.7 ± 0.0 Aa | 4.8 ± 0.1 Aa | 16 ± 0.4 Abc | 15.8 ± 0.3 Aab | 39.1 ± 0.5 Aa | 34.1 ± 0.5 Ba | 621 ± 15 Aa | 539 ± 12 Ba |

[x] Denotes the mean standard error. [y] Values followed by similar uppercase letters among season (column) within location (row) indicate no significant difference according to the Tukey mean test. [z] Values followed by similar lowercase letters among location (row) within season (column) indicate no significant difference according to the Tukey mean test.

**Table 7.** The interaction between sweet corn cultivars and seasons for ear diameter and ear length.

| Cultivar | Season | | | | | | | |
|---|---|---|---|---|---|---|---|---|
| | Spring | | Fall | | Spring | | Fall | |
| | Ear diameter | | | | Ear Length | | | |
| | cm | | | | cm | | | |
| Affection | 4.4 ± 0.1 [x] | A [y] bc [z] | 4.5 ± 0.1 | Aa | 17.5 ± 0.3 | Ae | 16.4 ± 0.3 | Bd |
| BSS1075 | 4.5 ± 0.1 | Aab | 4.5 ± 0.1 | Aab | 18.2 ± 0.1 | Aabcd | 16.5 ± 0.3 | Bcd |
| BSS8021 | 4.3 ± 0.1 | Ac | 4.1 ± 0.0 | Ad | 17.6 ± 0.2 | Ade | 17.0 ± 0.2 | Babc |
| Coastal | 4.7 ± 0.2 | Aab | 4.5 ± 0.1 | Babc | 18.6 ± 0.3 | Aab | 17.3 ± 0.3 | Ba |
| EX08767143 | 4.4 ± 0.1 | Abc | 4.5 ± 0.1 | Aab | 18.8 ± 0.2 | Aa | 16.7 ± 0.3 | Bbcd |
| Flagler | 4.4 ± 0.1 | Abc | 4.5 ± 0.1 | Aab | 18.6 ± 0.2 | Aab | 17.0 ± 0.2 | Bab |
| GSS1170 | 4.4 ± 0.1 | Abc | 4.4 ± 0.1 | Abc | 17.7 ± 0.2 | Acde | 16.9 ± 0.2 | Babcd |
| Obsession | 4.3 ± 0.1 | Abc | 4.4 ± 0.1 | Ac | 18.1 ± 0.3 | Abcde | 16.7 ± 0.3 | Bbcd |
| Passion | 4.4 ± 0.1 | Abc | 4.4 ± 0.1 | Abc | 18.4 ± 0.3 | Aabc | 16.8 ± 0.3 | Babcd |
| SCI336 | 4.3 ± 0.1 | Bbc | 4.5 ± 0.1 | Aa | 17.9 ± 0.3 | Acde | 16.7 ± 0.2 | Bbcd |

[x] Denotes the mean standard error. [y] Values followed by similar uppercase letters among season (column) within cultivar (row) indicate no significant difference according to the Tukey mean test. [z] Values followed by similar lowercase letters among cultivar (row) within season (column) indicate no significant difference according to the Tukey mean test.

For the main effect of cultivar on KTG (Table 5), cultivars SCI336 (671) and BSS1075 (651) had the highest KTG in southwest GA; cultivars Obsession (596) and BSS1075 (594) had the highest KTG in southeast GA; cultivar GSS1170 (602) had the highest KTG in south GA; cultivar Coastal (528) had the highest KTG in southwest AL; and cultivars Passion (640), EX08767143 (627), and SCI336 (620) had the highest KTG in central AL.

For the main effect of location on KTG within cultivars, southwest GA had the highest KTG within cultivars Affection (581), BSS1075 (651), EX08767143 (617), and SCI336 (671). Southeast GA had the highest KTG within cultivars BSS8021 (533), GSS1170 (588.0), and Obsession (596). South GA had the highest KTG within BSS8021 (526), Costal (571), Flagler (572), GSS1170 (602), and Obsession (546). Southwest AL had the highest KTG only within cultivar BSS8021 (465). Central AL had the highest KTG within cultivars BS8021 (511), Costal (576), EX08767143 (627), Obsession (581), and Passion (640).

In the interaction between location and season for ear quality parameters (Table 6), the main effect of location within season indicated that ear diameter means were similar between spring and fall seasons in southwest GA (4.3 and 4.4 cm, respectively) and central AL (4.7 and 4.8 cm, respectively). In southeast GA and southwest AL, ear diameter was higher in the spring (4.4 and 4.6 cm, respectively) compared to the fall season (4.2 and 4.3 cm, respectively); contrarily, ear diameter was higher in the fall (4.5 cm) compared to the spring (4.2 cm) in south GA. For the main effect of season within location, ear diameter was largest in central AL for both the spring and fall seasons.

The KRs were similar between spring and fall seasons for all locations, except in southwest GA where the KRs were larger in spring (17.7) compared to fall (16.3). Contrarily, the KIR was higher in the spring compared to the fall for all locations, except in southeast GA where KIR was statistically similar in the spring (33.7) and fall (34.2). The average KTG in the spring was higher than in the fall for all locations, except in southeast GA where the KTG were statistically similar between spring and fall seasons (552 and 544, respectively). For the main effect of season, individually, the largest KR in the spring was in southwest GA (17.7), and the largest KR in the fall was in south GA and southwest GA (16.6 and 16.3, respectively). The largest KIR in the spring was in central AL (39.0), and the largest KIR in the fall was in southeast GA and central AL (34.2 and 34.1, respectively). The KTG in the spring was the highest in southwest GA and central AL (632 and 621, respectively), and the largest KTG in the fall was in southeast GA (544), central AL (539), southwest GA (536), and south GA (528).

There was a significant interaction between cultivar and season for ear diameter and ear length (Table 7). For the main effect of cultivar within season, the ear diameter was

similar between spring and fall seasons for all cultivars, except for Coastal, for which ear diameter was higher in spring (4.7 cm) compared to the fall (4.5 cm), and cultivar SCI336, for which ear diameter was higher in the fall (4.5 cm) compared to the spring (4.3 cm). Ear length was higher in the spring compared to fall, regardless of cultivar. For the main effect of season within cultivar on ear diameter, the highest ear diameter in the spring was measure in cultivar Coastal (4.7 cm), but the lowest was in cultivar BSS8021 (4.3 cm). The highest ear diameter in the fall was measured in cultivars SCI336 (4.5 cm) and Affection (4.5 cm) and the lowest was in cultivar BSS8021 (4.1 cm). The longest ear length in the spring season was measured in cultivar EX08767143 (18.8 cm), while cultivar Costal (17.3 cm) had the longest ear length in the fall season. The cultivar Affection had the lowest ear length in both the spring and fall seasons (17.5 and 16.4 cm, respectively).

### 3.5. Multivariate and Correlation Analysis

For the PCA, cultivars within location and season were considered individually (Figure 2). For example, the cultivar Affection grown in the spring season of southwest GA was an individual. Individuals were clustered in two groups, with PC1 and PC2 explaining 58.1% of the total variance of the data. Most individuals were clustered together in the largest group (represented by the blue color in Figure 2a), which had the highest values for all variable responses, except for the number of ears per plant (EAR). The second cluster group (represented in red color in Figure 2a) had a lower number of individuals compared to the first cluster group. Particularly, the second cluster group had higher values of EAR compared to the first cluster group.

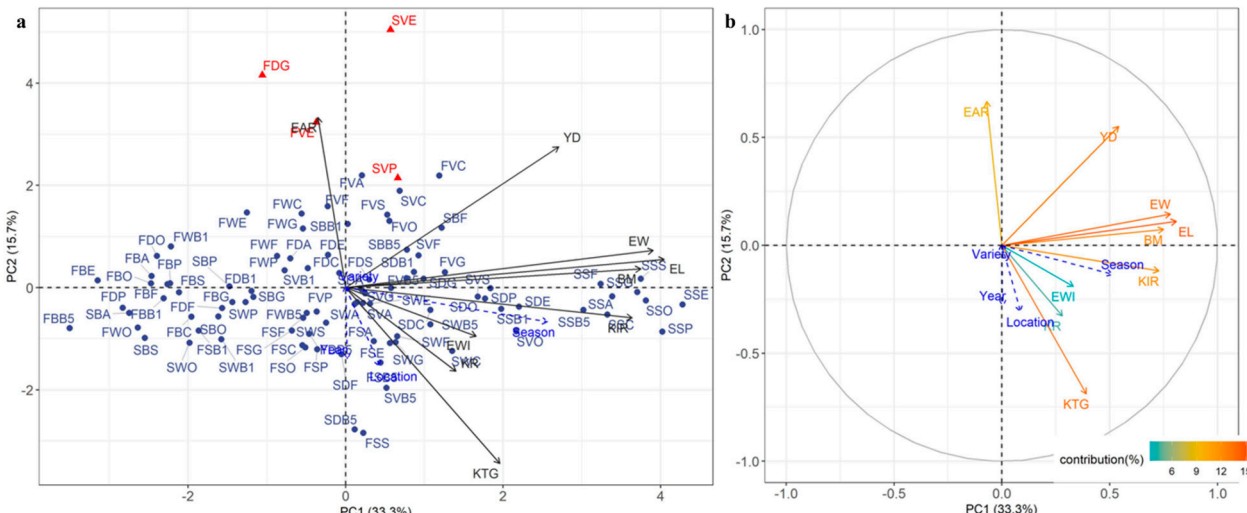

**Figure 2.** The principal component analysis (PCA) biplot is split into two graphics of all individuals and variables distribution and clustering (**a**) and variables correlation and contribution plot (**b**). Note: In (**a**), first letter indicates season (S = Spring, F = Fall); second letter indicates location (D = Southwest GA, W = South GA, V = Southeast GA, S = Central AL, B = Southwest AL); third and fourth letter indicates cultivars (A = Affection, B5 = BSS1075, B1 = BSS8021, C = Coastal, E = EX08767143, F = Flagler, G = GSS1170, O = Obsession, P = Passion, S = SC1336). In (**b**), EAR = number of ears, YD = yield, EW = ear weight, EL = ear length, BM = biomass, KIR = kernel grain in a row, EWI = ear width or diameter, KR = kernel row grain, KTG = kernel total grains.

In the variable correlation analysis (Figure 2b), variable responses were all clustered on the right side of the plot, except for EAR, indicating that yield, ear weight, biomass, ear length, and KIR are positively correlated. Variables KTG, KR, and ear width were negatively correlated with EAR. The quality of the response variables can be analyzed through the distance between them and the origin in the plot. Variables that are far away from the origin are well represented in the data; for instance, the EAR, yield, ear weight, ear length, biomass, KIR, and KTG are variables with the highest quality of response. The

contribution of the response variables is represented in percentage (%), where the "warmer" color represents a high percentage of contribution. For instance, the KR and ear width had a lower percentage of contribution; contrarily, yield, ear weight, ear length, and KTG had a higher percentage of contribution.

The Pearson's correlation analysis (Figure 3) indicated that sweet corn yield is positively correlated with ear weight and biomass. Similarly, ear length had a positive correlation with biomass, ear weight, and KIR. The KTG was positively correlated with KIR and KR; contrarily, the KTG has a negative correlation with EAR.

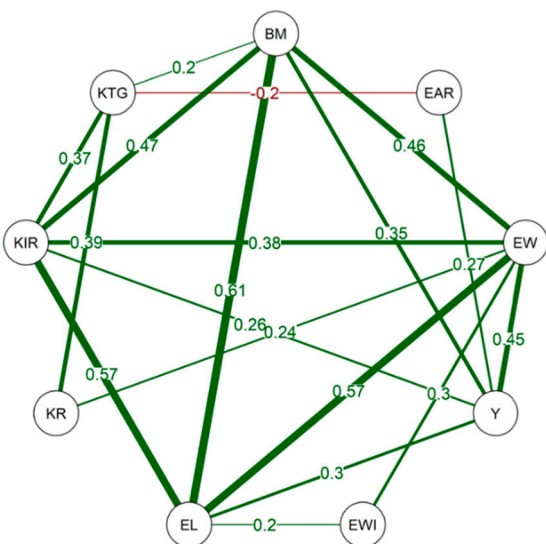

**Figure 3.** Correlation-based network analysis using Pearson's correlation method to compare all response variables, number of ears per plant (EAR), ear weight (EW), yield (Y), biomass (BM), ear diameter (EWI), ear length (EL), kernel rows (KR), kernel grains in a row (KIR), and kernel total grains (KIR).

## 4. Discussion

Climate change and weather variability have impacted the growth and development of vegetable crops worldwide [16–22]. In the southeastern U.S., heavy rainfall events, unpredictable heat and drought stress, and frequent high-temperature fluctuation reduce sweet corn crop development, resulting in decreased yields and quality [18,23–25]. The impact of the weather variability is further enhanced by the common use of super-sweet cultivars of sweet corn, which have the highest potential yield but are the most sensitive to drastic changes in daily air temperature and soil water availability [26]. Understanding the plant response to environmental conditions and selecting the most adaptable cultivar for subtropical environmental conditions is the first step in developing the best management practices for sweet corn production in the southeastern U.S.

Rainfall accumulations were similar across locations and seasons, matching the crop water requirements of 268 mm for sweet corn grown in the southeastern U.S. [27]. However, scattered heavy rainfall events caused soil water saturation conditions, creating anaerobic conditions that reduce root growth while inducing soil nutrient leaching [28]. Such heavy rainfall events occurred at 8 DAP (110 mm) in the spring and at 21 DAP (104 mm) in the fall of southwest GA; at 32 DAP (50 mm) in the spring and at 20 DAP (42 mm) in the fall of southeast GA; at 21 DAP (58 mm) and 70 DAP (68 mm) in the spring and 29 DAP (65 mm) in the fall of south GA; at 18 DAP (81 mm) and 32 DAP (59 mm) in the spring and at 24 DAP (65 mm) in the fall of southwest AL; and at 10 DAP (52 mm) and 21 DAP (65 mm) in the spring of central AL. Particularly, southwest AL was the location with the largest number of rainfall events and accumulated rain, which explains the lowest yield of sweet corn cultivars within that location for both growing seasons. The cultivar Coastal

stood out from the other cultivars in southwest AL and showed a high potential for good ear development and yield, even in waterlogged conditions.

Average daily air temperatures were also similar among locations and seasons and were within the optimal range for sweet corn production, which varies between 20 and 30 °C [23,24]. In general, daily air temperatures were lower than is optimal for sweet corn production in early spring, when sweet corn plants are in the vegetative stage, but daily air temperature increased and reached optimum levels during late spring, when sweet corn plants were undergoing ear development. Low daily air temperatures in the early spring reduced GDD accumulation in the vegetative stage. Consequently, there was an increase in sweet corn l that allowed for higher NM values during the spring compared to the fall. Ultimately, the extended growing season in the spring compared to the fall allowed for the highest sweet corn yields [29–31]. Particularly, daily air temperatures later in the spring increased and was within optimum levels during sweet corn reproductive stages for all locations, except in southeast GA, which had 20 days with daily air temperatures above 30 °C during the reproductive stages. Daily air temperatures above the optimum for sweet corn crop development decreased yield potential [15]; still, cultivars EX08767143 (26.7 Mg ha$^{-1}$), Coastal (23.7 Mg ha$^{-1}$), and SCI336 (23.5 Mg ha$^{-1}$) had the largest yields for that location, demonstrating their tolerance to heat stress.

During the fall season, daily air temperatures were higher in the early season and reduced with crop development. In response, there was a quick GDD accumulation that increased sweet corn k and shortened the period between planting and harvest. This negatively impacted ear diameter, KIR, KR, and KTG, because the poor biomass accumulated during the vegetative stage was not able to ensure grain filling during the reproductive stage [32]. Consequently, the shorter growing season of fall compared to spring resulted in the lowest sweet corn yields. Similar results were previously reported in cabbage production for the southeastern U.S., where high temperatures in early fall shortened the vegetative stage and reduced cabbage head size [33].

Overall, season and location were the main factors impacting sweet corn cultivar performance according to the PCA, which corroborates previous studies [23,24]. The cultivars with the best performance in the spring were Affection, GSS1170, Passion, and SCI336, and in the fall were Affection, GSS1170, and SC1136. The results also indicate that sweet corn yield is strongly correlated with ear width and ear length but poorly correlated with KTG, suggesting that breeding programs trying to increase the potential yield in sweet corn should be focused on ear dimensions instead of KR, KIR, and KTG.

## 5. Conclusions

Weather variability in the humid subtropical environmental conditions of the southeastern U.S. is impacting sweet corn production. Particularly, heavy rainfall events, unpredictable heat and drought stresses, and frequent high-temperature fluctuation create challenges during crop growing seasons. In this study, sweet corn cultivars were evaluated for five locations of the southeastern U.S. in the spring and fall. Daily air temperatures had a direct impact in sweet corn development, yield, and ear quality, while heavy rainfall events caused situations of waterlogging in all locations for both growing seasons. The results indicated that cultivar performance was more impacted by season than location. Low daily air temperatures in early spring delayed crop growth and allowed for larger biomass accumulation in the spring compared to the fall, when high daily air temperatures shortened the growing season. Sweet corn yields were thereby higher in the spring compared to the fall. Overall, heavy rainfall events negatively impacted sweet corn development, and the cultivars with great potential for resistance to environmental stresses and best performance for most locations were Affection, GSS1170, Passion, and SCI336 in the spring growing season, and Affection, GSS1170, and SC1136 in the fall growing season.

**Author Contributions:** Conceptualization, J.P., T.C. and A.L.B.R.d.S.; methodology, J.P. and A.L.B.R.d.S.; software, J.P. and A.L.B.R.d.S.; validation, T.C., W.F. and A.L.B.R.d.S.; formal analysis, T.C. and A.L.B.R.d.S.; investigation, J.P. and B.H.; resources, T.C. and A.L.B.R.d.S.; data curation, J.P. and A.L.B.R.d.S.; writing—original draft preparation, J.P.; writing—review and editing, W.F., T.C., M.S.-G., K.K. and A.L.B.R.d.S.; visualization, A.L.B.R.d.S.; supervision, A.L.B.R.d.S. project administration, A.L.B.R.d.S.; funding acquisition, A.L.B.R.d.S. All authors have read and agreed to the published version of the manuscript.

**Funding:** This research received no external funding.

**Acknowledgments:** Authors would like to thank farm managers for each location where field experiments were conducted, as well as members of the Vegetable Crop Team at Auburn University for helping with sampling and data collection.

**Conflicts of Interest:** The authors declare no conflict of interest.

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
