# Peer review of "Characterization of Sweet Corn Production in Subtropical Environmental Conditions"

_agriculture, doi:10.3390/agriculture13061156_

Round 1

Reviewer 1 Report

This study provides valuable insights into the performance of different sweet corn cultivars under the environmental conditions of the southeastern U.S. It emphasizes the importance of considering both weather variability and seasonal differences when making cultivar selections for sweet corn production in this region. Some minor issues need to be addressed before publication:

1. Some statistical numbers need to be added in the Abstract to further show why some cultivars should be grown in spring and some should be in winter.

2. Line 41: Changes in rainfall patters also have been creating challenges. Please consider the following ref to further support this statement.

Liu, K., Harrison, M.T., Yan, H. et al. Silver lining to a climate crisis in multiple prospects for alleviating crop waterlogging under future climates. Nat Commun 14, 765 (2023). https://doi.org/10.1038/s41467-023-36129-4

3. Table 3 should add some statistical analysis e.g. ANOVA

Author Response

Authors appreciate the time and expertise of the reviewer in reviewing our manuscript. We carefully incorporated all suggestions which have significantly improved the readability of the paper. The attached file contains all responses to the reviewer's comment.

Reviewer 2 Report

In this article, the authors evaluated and characterized the performance of ten commercial sweet corn cultivars exposed to several environmental conditions of southeastern U.S. and described the impacts of weather variability on cultivar development, yield, and ear quality. The topic is interesting and write very well. Only some minor points need to be corrected before publication.
1. For Tables 3-7, the error value should be provided.
2. Abbreviation should be used throughout the manuscript.
3. Please check the structure of format presentation for Agriculture.

Minor editing of English language required

Author Response

(The authors gave the same response as above.)

Reviewer 3 Report

This manuscript used five field experiments in southeastern US to evaluate and characterize the performance of ten commercial sweet corn cultivars exposed to several environmental conditions. The manuscript is well written. The results have a good practical significance in southeastern US. I would suggest this paper for publication in Agriculture after clearly define the scope of the conclusion. The most important conclusion is cultivar selection should be performed according to season rather than location. The conclusion should emphasize the environmental condition (humid subtropical climate), if you want to say which variety is better.  

Author Response

(The authors gave the same response as above.)

Round 2

Reviewer 2 Report

no